# Multielectrode Arrays at Wafer-Level for Miniaturized Sensors Applications: Electrochemical Growth of Ag/AgCl Reference Electrodes

**DOI:** 10.3390/s23136130

**Published:** 2023-07-04

**Authors:** Haosheng Wu, Robert Krause, Eshanee Gogoi, André Reck, Alexander Graf, Marcus Wislicenus, Olaf R. Hild, Conrad Guhl

**Affiliations:** 1Fraunhofer Institute for Photonic Microsystems (IPMS), Center Nanoelectronic Technologies (CNT), An der Bartlake 5, 01109 Dresden, Germany; robert.krause@ipms.fraunhofer.de (R.K.); andre.reck@ipms.fraunhofer.de (A.R.); marcus.wislicenus@ipms.fraunhofer.de (M.W.); conrad.guhl@ipms.fraunhofer.de (C.G.); 2Fraunhofer Institute for Photonic Microsystems (IPMS), Maria-Reiche-Str. 2, 01109 Dresden, Germany; alexander.graf@ipms.fraunhofer.de (A.G.); olaf.hild@ipms.fraunhofer.de (O.R.H.)

**Keywords:** chlorination, reference electrodes, silver/silver chloride, underlying metal layers

## Abstract

In this study, a range of miniaturized Ag/AgCl reference electrodes with various layouts were successfully fabricated on wafer-level silicon-based substrates with metallic intermediate layers by precisely controlling the electrochemical deposition of Ag, followed by electrochemical chlorination of the deposited Ag layer. The structure, as well as the chemical composition of the electrode, were characterized with SEM & EDS. The results showed that the chlorination is very sensitive to the applied electric field and background solution. Potentiostatic chlorination, in combination with an adjusted mushroom-shaped Ag sealing deposition, enabled the formation of electrochemical usable Ag/AgCl layers. The stability of the electrodes was tested using open circuit potential (OCP) measurement. The results showed that the reference electrodes stayed stable for 300 s under 3 M KCl solution. The first stage study showed that the stability of the Ag/AgCl reference electrode in a chip highly depends on chip size design, chlorination conditions, and a further protection layer.

## 1. Introduction

Miniaturization is the essential direction for electrochemical sensor development. A significant reduction in electrochemical sensor size extends their applications in various fields, such as trace element determination [1,2] and lab on chip analytics for fast chemical/biological analytics [3]. During the fabrication of miniaturized electrodes, multielectrode arrays improve material efficiency. For some applications, multielectrode arrays are even indispensable, such as collecting enough neural signals for electronic circuitry in medical applications [4], and microelectrode arrays of photogalvanic cells for higher solar energy conversion efficiency [5].

Wafer-level fabrication, i.e., microelectronic process technologies, includes three key process steps: photolithography, deposition, and etching. These technologies enable various electrode layout designs on a broad range of material layers with high precision and reproducibility. Wafer-level fabrication also enables the integration of microelectrodes of various shapes and sizes in a small area with defined distance between each other, which is scalable over several orders of magnitude. Compared to the traditional miniaturized electrodes fabrication techniques, such as screen-printing, wafer-level fabrication provides more alternatives in electrode materials, substrate materials, and layout designs from a larger scale (down to nanometer scale), which thus enhances their application possibilities.

Challenges also exist in the wafer-level fabrication of various cell designs, as it potentially affects the process results for microelectronic production techniques. That is, for reproducible electrochemical analysis, an accurate electrode arrangement is just as important as pure electrode materials and precise knowledge of the electrode sizes. A Ag/AgCl reference electrode was chosen in this study due to its environmentally friendly quality, better potential stability, and the high sensibility of chloride, which enables a wide range of applications [6,7,8,9]. However, the fabrication of Ag/AgCl reference electrodes is also one of the most critical issues in the design of miniaturized electrochemical sensors. Until now, no direct studies of Ag/AgCl reference electrodes on wafer-level multielectrode arrays have been conducted.

The aim of our study is thus to investigate the deposition of Ag and AgCl layers utilizing electrochemical deposition on a silicon-based substrate with a complex underlying metal layer stack, as well as multiple electrode layouts. That is, to investigate the electrochemical deposition of a metallic Ag layer and the subsequent chlorination of Ag and analyze the performance of the Ag/AgCl reference electrodes.

## 2. Materials and Methods

A high number of electrochemical measurement cells were fabricated on a 6-inch wafer. They were divided into 6 chips, sorted by their electrode geometry, size, and distance between electrodes. Each cell contains a working electrode (WE), a counter electrode (CE), and a reference electrode (RE).

Figure 1 demonstrates chip 1 and chip 6 with the location of cell XS and cell XL, respectively. Cell XS and cell XL have the largest geometric difference in size, shape, and electrode distance and were thus chosen as candidates for analysis throughout the whole study. The measurement cell XS, with an electrode diameter of 20 µm and an electrode distance of 50 µm, is the smallest measurement cell among all, while the measurement cell XL, with an electrode diameter of 100 µm and an electrode distance of 200 µm, is the largest measurement cell. Figure 1 illustrates the geometric shape of the three electrodes. (for more information on chip design and layout see O. R. Hild et al. [10]).

### 2.1. Galvanostatic Metallization of Ag

The pre-structured areas in different modules were designed with the same metal layer stack as shown in Figure 2. A definition is made in the process development with a feedback loop to the layout to consider material-specific boundary conditions, e.g., conductivity, layer thicknesses, etc.

The silicon substrate of the wafer is coated with silicon oxide as an insulating layer. AlSiCu interconnects were deposited for connecting the electrode areas. A 4 µm nickel barrier layer was deposited further on the interconnect, followed by a thin Au seed layer (target thickness 100 nm), which was accomplished by the electroless process technology. Gold was chosen as the seed layer due to its excellent adhesive force to nickel and silver layers. The nickel barrier layer was necessary to overcome the rapid formation of brittle aluminum-gold intermetallic at high temperatures [11,12,13].

An NB Semiplate Ag 100 (NB Technologies GmbH, Bremen, Germany) was used for the deposition of silver. The current source used was a Potentiostat/Galvanostat VSP3 and a Booster VMP3 (Fa. Bio-Logic GmbH, Göttingen, Germany). The reference electrode (RE) was deposited with silver with a current density of 15 mA/cm^2^ at a bath temperature of 50 °C. The target thickness of silver was calculated based on the Faraday law:$$h_{Ag}=\varepsilon \frac{MIt}{A\rho zF}$$
where $h_{Ag}$ is the thickness of the Ag layer;  z valency of the Ag ion;  ε is the deposition efficiency (0.625 was taken); I is the applied current; A is the open surface area; M is the molecular weight of Ag; *t* is the total time of electrochemical deposition; *F* is the Faraday constant; ρ is the density of Ag.

### 2.2. Electrochemical Chlorination of Ag

Figure 3 shows the setup for the galvanostatic and potentiostatic chlorination, which were both applied in this study. The chlorination took place in a beaker. During the chlorination, the 6-inch wafer was diced into modules to better fit into the beaker.

#### 2.2.1. Galvanostatic Chlorination

Galvanostatic chlorination was first chosen for AgCl layer generation at 25 °C. Chlorination was carried out under current densities of 7.5 mA/cm^2^, 15 mA/cm^2^, and 20 mA/cm^2^ in the same background solution of 0.1 M HCl, respectively. Based on the results, chlorination under 7.5 mA/cm^2^ with a background solution of 0.1M HCl at pH 1, 0.01 M HCl at pH 2, 0.1 M KCl at pH 6.8, and 1 M KCl at pH 6.8 were investigated further.

#### 2.2.2. Potentiostatic Chlorination

Potentiostatic chlorination was further chosen for better control over occurring reactions by potential limitation, ensuring fabrication repeatability of the various geometric chip design and better electrode performance. Chlorination with a background solution of 1 M KCl at pH 6.8 at 25 °C was carried out at a constant potential of 0.2 V for 40 s, 0.5 V for 40 s, 0.75 V for 40 s, 1.0 V for 10 s, and 1.0 V for 40 s, respectively.

### 2.3. Characterization of Ag/AgCl Reference Electrode

Analytical characterization of the electrodes using scanning electron microscopy (SEM) and energy dispersive X-ray spectroscopy (EDS) was carried out to evaluate the deposition process result of the different electrode layers, as well as the chlorination of Ag. Furthermore, any possible diffusion of the Cl towards the underlying layers would be revealed. The SEM analysis of the electrodes was executed using an electron microscope of the type ApreoS (Fa. Thermo Fisher Scientific, Eindhoven, The Netherlands) with an attached EDS detector of the type X-Flash 6/60 (Fa. Bruker, Nano GmbH, Berlin, Germany). The software version used for the element analysis was Esprit 2.3.

For high-resolution SEM and EDS of the cross sections, selected cells were prepared by means of a focused ion beam (FIB). Characterization of the cell surface (deposition quality, grain size evaluation) with the deposited AgCl layer was performed beforehand. The utilized acceleration voltages of the SEM were 2 kV and 5 kV each, at a beam current of 25 pA, in conjunction with secondary electron detectors (for topography contrast), as well as backscattered electron detectors (for material contrast). The element analysis was only carried out on the FIB cross-sections with higher voltages of 20 kV in order to generate sufficient electron signals. Elemental mappings of the different electrode layers, as well as line scans, were part of the characterization as well.

### 2.4. Open Circuit Potential (OCP) Measurement

The open circuit potential (OCP) measurement was carried out for a performance test of the fabricated Ag/AgCl electrodes under the three potentiostatic chlorination conditions at 20 °C. During the OCP measurements, the 6-inch wafer was diced into modules, which contained a total 18 chips, including cell XS and cell XL. As shown in Figure 4, a conventional macroscopic Ag/AgCl reference electrode was alternately used as the counter electrode (CE) of the electrochemical system to compare their performances. A 3 M KCl electrolyte solution was chosen as the background solution, since it is also the standard background solution of the commercial Ag/AgCl reference electrode.

## 3. Results and Discussion

### 3.1. Galvanostatic Ag Chlorination on Ag Layer

Silver anodization under galvanostatic conditions is frequently used for the fabrication of AgCl on top of the Ag layer [14,15,16]. Even though the Ag deposition for each cell was dense and the Ag layers held a similar depth which is parallel to the surrounding silicon dioxide, different chlorination results of electrodes can be seen in Figure 5.

For all the electrodes treated under 15 mA/cm^2^ and 20 mA/cm^2^, the Ag layer and the underlying metal layers were anodized, and electrodes suffered from corrosion. Electrodes contained deep cracks throughout, as shown in Figure 5d, or the whole structure turned into empty holes, leaving only the silicon oxide substrate underneath as shown in Figure 5c, indicating an even stronger corrosion which led to complete dissolution or break out of the metal layers. Under a current density of 7.5 mA/cm^2^, some electrodes resulted in the successful growth of AgCl on the Ag layer without anodization of other layers, as shown in Figure 5b, but electrodes with corrosion effects, as shown in Figure 5d, cannot always be avoided.

Figure 6 shows the top-down SE images combined with element mapping via EDS, presenting the elemental distribution over the electrode. The cracked amorphous top layer contains Ni, Al, O, and Cl. Plausible origins of these elements are the electrolyte (Cl), the barrier layer (Ni), and the underlying interconnect material (Al). The presence of these elements indicates deep corrosion throughout the whole cell.

#### 3.1.1. Corrosion Effect of Underlying Metal Layers

Based on Ohm’s law, it is well known that for maintaining constant current, the potential would increase with the growth of the AgCl film on the surface of Ag as the resistance increases. The transformation of Ag to AgCl itself is also a volume expansion procedure. These issues becomes critical when it comes to the micrometer-level substrate with multiple underlying metal layers. Such a corrosion effect was also mentioned by Brewer [16,17]. He reported that electrodes with different geometric surface areas suffered from transient potential increases under constant current, which was believed to stress the electrodes strongly. In our study, a constant current density between each electrode was also doubted, since the growth of the AgCl layer on the Ag surface was sensitive to geometric limits and underwent an intense potential change. Thus, under a constant current, not only was the potential unpredictable, but the transient current density change was too. A potentiostatic chlorination was thus chosen for further approaches.

Furthermore, comparing the results under all the background solutions, a severer corrosion effect was reached under acidic conditions due to the much higher conductivity with the increased concentration of H^+^ (Grotthus mechanism), as well as increased generation of H_2_ gas. It is thus worth mentioning that, even though 0.1 M HCl is often reported to be the standard background solution for silver chlorination, for miniaturized electrodes with complex, intermediate, less novel metallic layers, it is challenging to use. For further potentiostatic chlorinations, 1 M KCl was chosen as the background solution.

#### 3.1.2. Improvement Using Mushroom-Shaped Ag Electrochemical Deposition

The first design of the Ag plating was a filling up to the surface level of the surrounding silicon oxide (see Figure 5a). As discussed in the former section, corrosion of the underlying, less noble metals is a serious issue preventing the successful formation of miniaturized on-chip electrodes. Penetration of corrosive species (e.g., Cl^−^) along the Ag/SiO_2_ interface was deemed to be a cause for the observed effects. To prevent such unfavorable effects, Ag overplating was used to seal the Ag/SiO_2_ interface area.

As shown in Figure 7c,d, depending on the reference electrode size of the geometric area on each cell, the deposition height over the surrounding silicon dioxide varied from 2.4 µm to 5.2 µm, reaching a total Ag deposition between 5 µm to 10 µm. It can be seen in Figure 7 that dense Ag grains were successfully grown with a homogenous, one-phase structure.

### 3.2. Potentiostatic Ag Chlorination on Ag Layer

The AgCl layer was successfully deposited on each electrode under all the conditions (0.2 V, 0.5 V, and 1.0 V).

Under the conditions of 0.2 V and 40 s, the surface of the reference electrode in cell XS was covered completely and densely with angular AgCl crystallites, while there were still Ag grains explored on the surface in cell XL. As can be seen in Figure 8a,b, compact AgCl grains were grown on cell XS with no damage or pores on the surface. However, the cross-section BSE image of cell XS shown in Figure 9a revealed that only a very thin layer of AgCl was deposited on the Ag layer, with 438 nm on cell XL and 365 nm on cell XS. The Ag/Cl ratio on cell XL was approximately 3:1, while the Ag/Cl ratio on cell XS was around 2.5:1. It should also be noted that, due to the nanometer-thin and uneven layer of AgCl, the elemental ratio based on EDS contains large deviations as its information depth is larger than the AgCl thickness, which resulted in much higher Ag signal. Longer chlorination time is needed for proper chlorination under 0.2 V.

A much more homogenous AgCl deposition was reached under the conditions of 0.5 V 40 s. Both the surfaces of cell XS and cell XL were fully covered by angular AgCl crystallites, with Ag/Cl ratios close to 1:1; also, no large grain size and deposition depth differences between the AgCl layers on cell XS and cell XL can be seen, pointing towards a large process window for the implementation of various cell layouts. It has been noticed that a three-region Ag/AgCl layer was generated. As shown in Figure 9b, the Ag/AgCl layer on top was a compact AgCl layer, followed by a porous layer containing both AgCl and Ag in between, and a dense Ag layer at the bottom. Furthermore, microchannels in the micrometer range and nanopores with sizes around 100 nm existed over the AgCl surface on each cell, as illustrated in Figure 9c,d.

Under the condition of 1 V, the microchannel got longer and pores were more abundant and slightly bigger between the AgCl grains, as can be seen in Figure 8e,f. However, instead of a porous middle layer, a hollow gap between the AgCl and Ag layer was generated, as shown in Figure 9c,d. Under the conditions of 1 V 40 s, most of the Ag layer in cell XL was completely anodized, with only AgCl left on the surface, leaving a hollow gap between the AgCl layer and the Au seed layer, as shown in Figure 9c, while cell XS suffered the corrosion effect and no proper AgCl layer could be seen.

#### Growth Mechanism and Reaction of AgCl Growth

By comparing the top-down images in Figure 8 and the cross-section images in Figure 9, conclusions on the growth mechanism of AgCl on the surface of Ag and subprocesses of Ag chlorination can be drawn. Based on the fact that a thin porous layer containing mixtures of Ag and AgCl was generated under lower potential (0.2 V), it is possible to hypothesize that the starting point of AgCl nucleation tends not to take place directly on the surface of Ag, which created spaces between the surface of Ag and the AgCl nuclei, and thus created, as the first layer, a macroscopically porous microstructure, shown in Figure 10a. The same observation was also reported by Lou [18]: that the nucleation and growth of AgCl tended to not be directly on the Ag surface but rather just close to the Ag surface in the solution. It was also mentioned that with stirring of the solution, as is the case in our study, more irregular and smaller AgCl crystallites were generated. It seemed that the growth on such a non-polished Ag surface was always irregular, as no clear orientation could be seen on the porous structure. No orientation preferred on the nucleation layer was also mentioned by Katan [19]. The roughness of the Ag surface might also play an important role, as the AgCl nuclei tend to aggregate at grooves or places where they are less affected by ion diffusion, which also led to porous structures.

As shown in Figure 9b,d, the growth of AgCl was layer by layer, which was also reported by other studies [20]. More interesting is that a compact AgCl layer on top of the porous AgCl nucleation layer was generated with increased potential, or it could be the same growth process under constant potential over time. It indicates the Cl^−^ ions can diffuse in the dense layer, allowing for further AgCl crystalline growth. As shown in Figure 10, the nucleation sites were not saturated at the first stage, at which the number of nuclei increased with the rather limited growth rate of each nucleus. In the second stage, the AgCl nuclei were saturated and sufficient Cl^−^ ions amount enabled the increasing of the growth rate of each nucleus. AgCl grains grew thus into larger crystals, leaving a macroscopically compact structure. It also indicates that two stages of the AgCl growth simultaneously occurred at different AgCl layers, once the first porous AgCl layer was generated.

Moreover, the combination of the top-down image in Figure 8d and the cross-section image in Figure 9b contributes to an understanding of the distribution of nanopores and microchannels in the compact AgCl layer. Microchannels were only formed rather near the top surface, but nanopores were widely distributed in the whole compact AgCl layer. The increased anodization rate of Ag at higher potentials also enhanced the possibility of gas generation (H_2_), and the thickness of the irregular formation of the AgCl nucleation layer increased. Spaces between Ag and AgCl crystallites tended to be larger and larger, which led to a macroscopic visible hollow gap between AgCl and Ag as shown in Figure 9d.

### 3.3. Stability of Various Ag/AgCl Electrode Structures

To test the stability of the reference electrodes on the chips, open circuit potential (OCP) measurements of reference electrodes chlorinated under different solution conditions against a conventional Ag/AgCl electrode (with 3 M KCl as the inner electrolyte) in a 3 M KCl solution were carried out. As the results in Figure 11 show, the chip with reference electrodes chlorinated under 1 V 10 s lasted longest, with 300 s stability, followed by a less stable chip with 225 s stability which contained reference electrodes chlorinated under 0.5 V 40 s. It is worth noting that the electrodes chlorinated under 0.2 V 40 s can still last one minute, even though there were only a few hundred nanometers of porous layer containing a mixture of Ag and AgCl crystallites. A relatively large offset of its potential against the conventional Ag/AgCl electrode is thus due to its improper chemical composition.

#### 3.3.1. The Influence of Nanopores and Irregular Formation of the First Ag Nucleation Layer on the Stability of the Ag|AgCl Electrode

The issue of nanopores is reported by many researchers [18,19,20], notably that although their existence enhances the sufficient diffusion of interfering ions on the AgCl surface and thus results in a highly reproducible and stable reference potential, as is also the case in our study, it leads to the transformation of AgCl grains to other more soluble products (e.g., Ag_2_O), thus quickening the dissolution of the AgCl grains, which is crucial for miniaturized electrodes. Additionally, Cl^−^ can rapidly reach the underlying substrate materials; this caused further corrosion in the present case of standard semiconductor interconnects underneath the electrodes.

The relatively short stability time of all the reference electrodes in the 3 M KCl solution is deemed to be mainly due to the existence of nanopores in the whole structure within the compact AgCl layer, which led to the rapid diffusion of Cl^−^, reaching the underneath layers.

However, the decisive issue remains the porous AgCl layer existing for chips chlorinated under 0.5 V and the hollow gaps existing underneath the AgCl layer in the chip chlorinated under 1 V, which induces direct contact of high concentration Cl^−^ with underlying, less noble metal layers. Thus, the stability time remained short, within 5 min.

#### 3.3.2. The Influence of Grain Size and Chip Size Effect on the Stability of the Ag/AgCl Electrode

It is also natural to stress the importance of the AgCl grain size on long-term stability, as the larger the grain size, the more stable the AgCl layer is against external pressure. Due to the practical challenges, there was no separated OCP measurement test for cell XS and cell XL under each chlorination condition. However, it can be seen in Figure 12 that the duration of stability time of the reference electrode still holds a strong relationship with AgCl grain size, as the larger the grain size in general in cell XS and cell XL, the longer the stability time is, regardless of the underneath porous AgCl layer and the existence of hollow gaps. That is, for long-term stability, the grain size played a much more important role. For further performance improvements of the reference electrode, it is thus more important to maintain or even increase the AgCl grain size.

Furthermore, the results from the two candidates, cell XS and cell XL, in the chip have shown that it is very challenging to reach an ideal Ag chlorination for all the cells with different sizes without adding any reagents, as the results in Figure 8 and Figure 9 show. The large difference between the cells also led to a shorter stability time during the OCP measurement. The chip size thus also has an impact on the stability.

#### 3.3.3. Potential Development Based on the OCP Measurement Results

Based on the issue discussed above, a couple of developments can be taken into account to improve long-term stability. Regarding the chip size effect, the use of stepwise chlorination or organic additives for balancing the current density on the chip for the production of an AgCl reference electrode should be further considered. Potential improvements also include adding organic additives for the generation of an orientation-preferred AgCl nucleation layer; furthermore, a stepwise chlorination, with the first stage under static conditions without stirring, on a smooth and contamination-free Ag surface should be taken into consideration to reduce the creation of porous structures and increase the grain size. Regarding the intermediate metallic layer, a seed layer improvement of the electroless nickel layer deposition is also possible [11]. Furthermore, a protection layer such as agar [14] or polyamide [15] was also reported to significantly improve the stability.

## 4. Conclusions

Ag/AgCl reference electrodes with various geometric sizes between Ø 20 µm and Ø 200 µm as large-scale multielectrode arrays at the wafer-level with metallic intermediate layers were successfully fabricated. The most commonly used galvanostatic chlorination was proven to not be suitable for delicate microstructures with underlying standard semiconductor metal layers. The usage of potentiostatic chlorination, in combination with an adjusted mushroom-shaped Ag sealing deposition, enabled the formation of electrochemical usable Ag/AgCl layers on top of less noble standard interconnect metals.

By comparison of the cross-section and the top-down SE images of Ag/AgCl reference electrodes on different cells, a potential growth process of AgCl grains on an Ag substrate is developed. This supported the proposal of Lou [18] that AgCl growth occurs not directly on the Ag surface, proving instead to be a layer-by-layer growth. The cause of the porous AgCl layer was believed to be the irregular (no orientation preferred) formation of AgCl nuclei, which are not found directly on the Ag surface but tend to aggregate at less diffusion-affected areas, grooves, etc. Open circuit potential (OCP) measurement results revealed that the stability of the reference electrode is limited to approximately 5 min in a 3 M KCl solution, due to both the existence of nanopores in the compact AgCl layer and the underlying porous AgCl layer. It is almost inevitable to have nanopores during Ag chlorination. The generation of an underlying porous AgCl layer could be potentially avoided by further improvements.

In summary, this first stage study reveals the possibility of Ag/AgCl fabrication directly on chip substrates, including standard semiconductor interconnects. The long-term stability of the Ag/AgCl reference electrode in the chip depends highly on the chlorination conditions, a further protection layer, as well as chip size design.

## Figures and Tables

**Figure 1 sensors-23-06130-f001:**
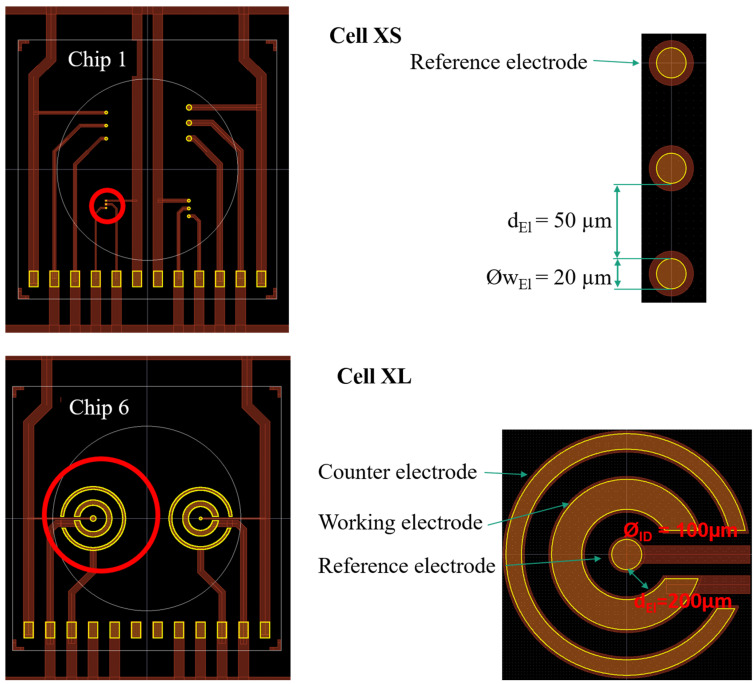
Overview of chips 1 and 6, with chip 1 containing the smallest cell, cell XS, and chip 6 containing the largest cell, cell XL. Detailed illustration of cell XS and cell XL showing the position of the reference electrode (RE), the working electrode (WE), and the counter electrode (CE) of each cell.

**Figure 2 sensors-23-06130-f002:**
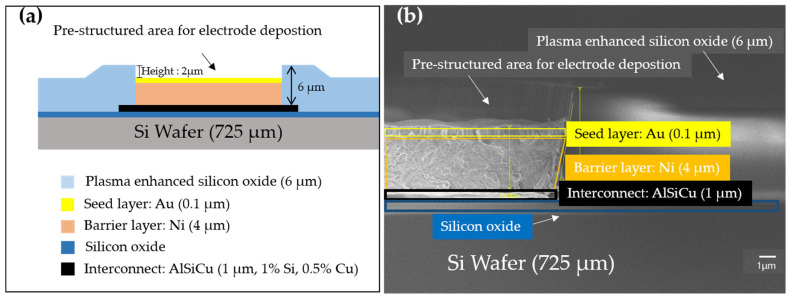
(**a**) Cross-section design of the layer structure, interconnects are to be produced with CMOS-compatible metals AlSiCu. (**b**) SEM cross-section image of the fabricated chip 4 as one example. The deviation of the thickness of each layer compared to the target design thickness is within 0.5%.

**Figure 3 sensors-23-06130-f003:**
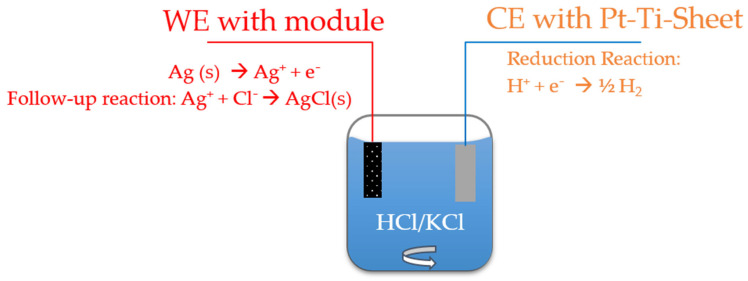
Schematic representation of the electrochemical chlorination setup.

**Figure 4 sensors-23-06130-f004:**
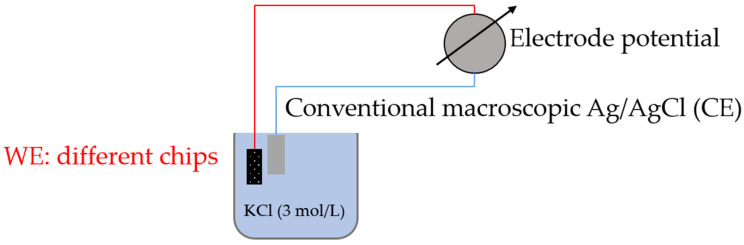
Schematic representation of the open circuit potential (OCP) measurement setup.

**Figure 5 sensors-23-06130-f005:**
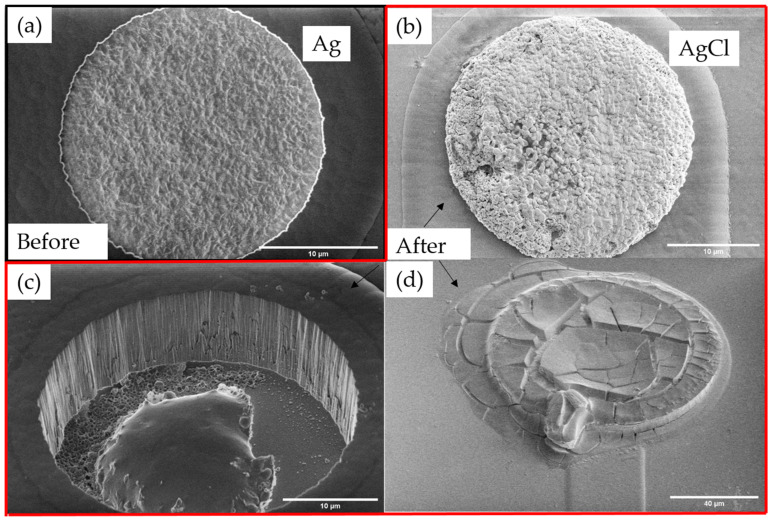
Top-down secondary electron (SE) images of Ag electrode (**a**) before galvanostatic chlorination, (**b**–**d**) after galvanostatic chlorination. Different deposition results occurred including (**b**) generation of AgCl layer, (**c**) corrosion with almost complete empty open area, (**d**) corrosion with cracks.

**Figure 6 sensors-23-06130-f006:**
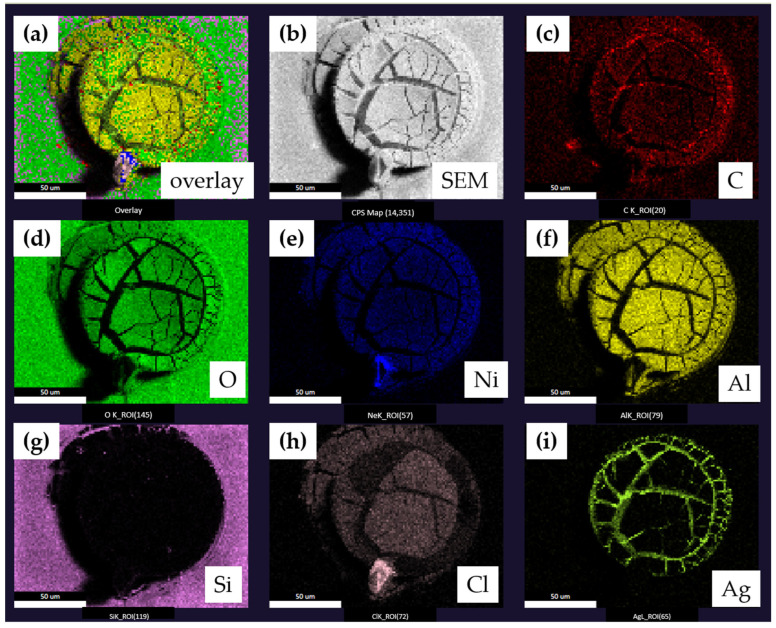
(**a**) Overlay images of SEM &EDS revealing all the elemental distribution of the corroded electrode; (**b**) top-down secondary electron (SE) image and (**c**–**i**) C, O, Ni, Al, Si, Cl and Ag element mapping by EDS of the corroded electrode, respectively.

**Figure 7 sensors-23-06130-f007:**
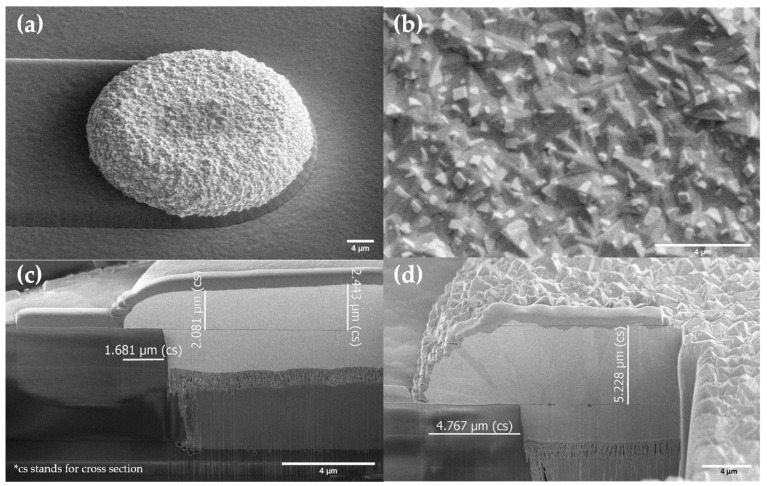
SEM Top-down and cross-section images of Ag deposition at the reference electrode position. (**a**) Top-down secondary electron (SE) image of cell XS; (**b**) close-up of top-down secondary electron (SE) image of cell XS; (**c**) cross-section SE image of cell XL; (**d**) cross-section SE image of cell XS.

**Figure 8 sensors-23-06130-f008:**
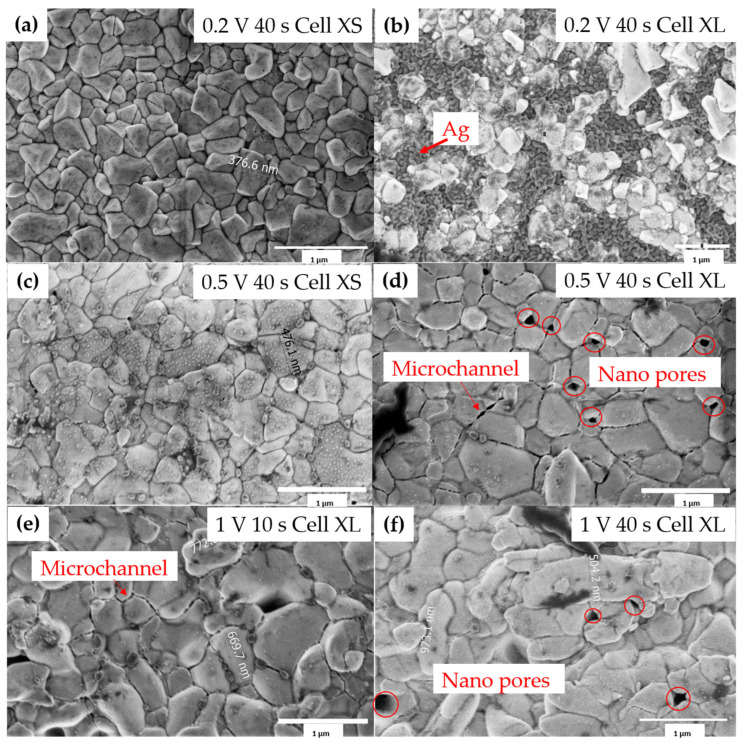
Top-down secondary electron (SE) images of the AgCl grains deposited under the condition of (**a**) 0.2 V 40 s for cell XS (**b**) 0.2 V 40 s for cell XL, (**c**) 0.5 V 40 s for cell XS, (**d**) 0.5 V 40 s for cell XL, (**e**) 1 V 10 s for cell XL, and (**f**) 1 V 40 s for cell XL. Note: under 1 V 40 s, cell XS suffered the corrosion effect, and no proper AgCl layer could be seen.

**Figure 9 sensors-23-06130-f009:**
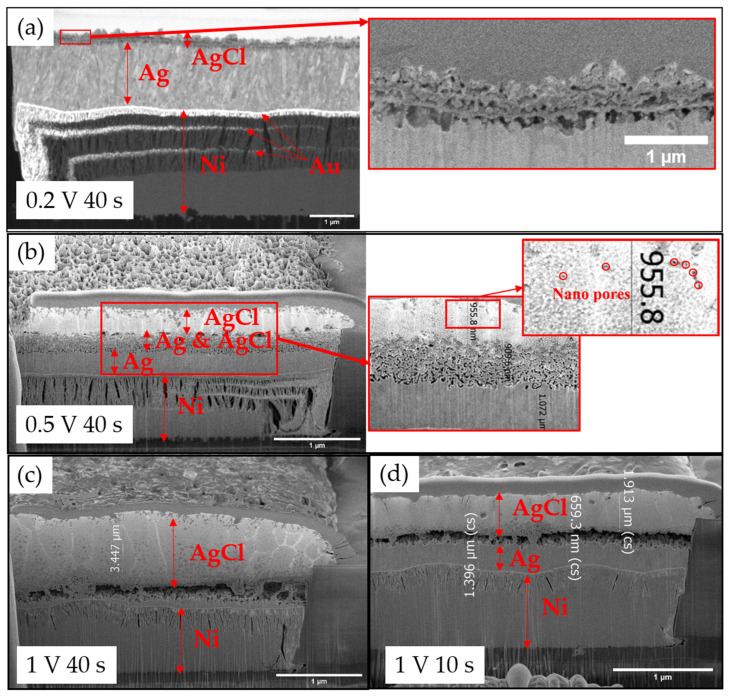
Cross-section SEM images of the AgCl grains deposited under the conditions of (**a**) 0.2 V 40 s for cell XS (BSE image), (**b**) 0.5 V 40 s for cell XL (SE image), (**c**) 1 V 40 s for cell XL (SE image), and (**d**) 1 V 40 s for cell XL (SE image).

**Figure 10 sensors-23-06130-f010:**
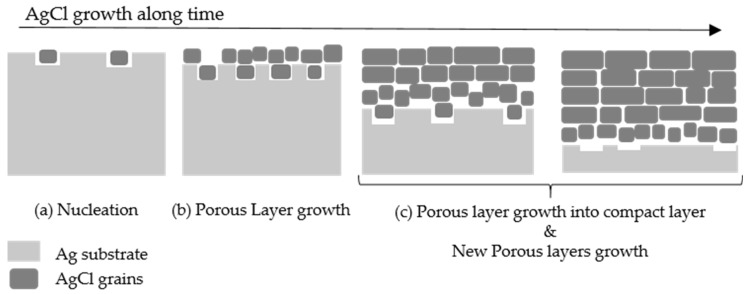
The potential growth process of AgCl on Ag substrate over time.

**Figure 11 sensors-23-06130-f011:**
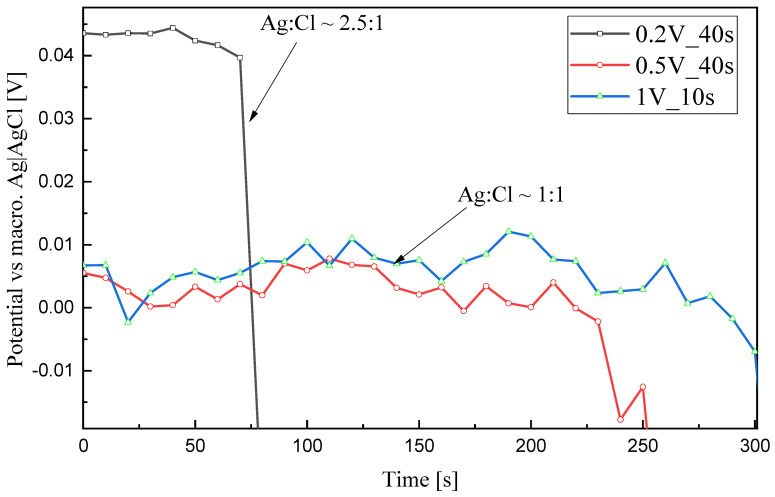
Potential of reference electrodes chlorinated under different conditions, which was measured against a conventional Ag|AgCl electrode in a 3 M KCl solution.

**Figure 12 sensors-23-06130-f012:**
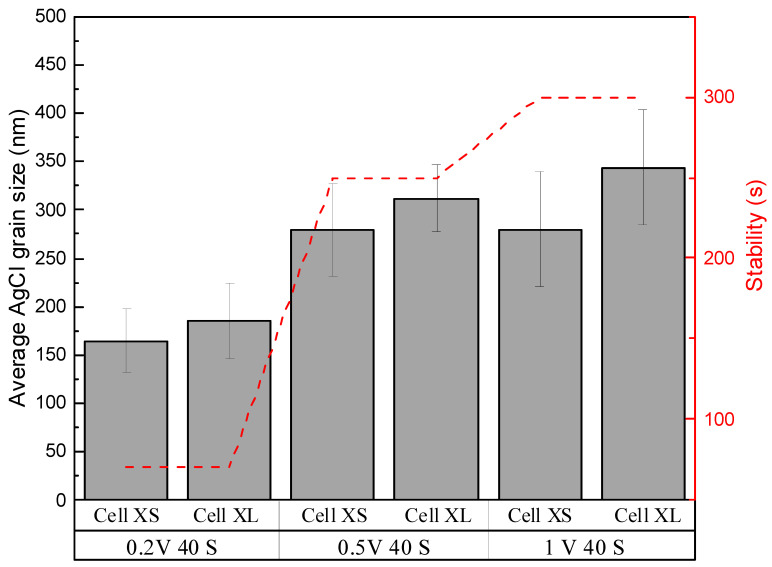
The relationship between the duration of the stability time of the chip (containing both cell XS and cell XL) during the OCP measurement and the average AgCl grain size in cell XS and cell XL chlorinated under 0.2 V 40 s, 0.5V 40 s, and 1 V 40 s.

## Data Availability

Data available on request due to restrictions.

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
