# Peer review of "Multielectrode Arrays at Wafer-Level for Miniaturized Sensors Applications: Electrochemical Growth of Ag/AgCl Reference Electrodes"

_sensors, 2023, doi:10.3390/s23136130_

Round 1

Reviewer 1 Report

Reviewer’s questions and recommendations are listed below.

Introduction should be improved with more detailed examples of microelectrode array analytical application. The importance of Ag/AgCl electrode fabrication instead of Ag (refer to screen printed electrodes) should be formulated more obvious.

According to Section 2.2, the two-electrode electrochemical cell was used in chlorination experiments. Why is it not the three-electrode electrochemical cell? Especially in potentiostatic mode, to be sure the measured potential is equal to desirable (applied).

The study looks incomplete. Despite the authors mentioned that it is a first stage of investigation, the manuscript suffers from the lack of electrochemical data and/or experiments with improves that the authors mentioned in text (lines 333-343). As suggestion: open circuit potential measurement should be enhanced with data and discussion for unmodified Ag layer and for solutions with lower KCl concentration or with another electrolyte (nitrate, sulphate, phosphate …), since chlorides diffusion is named one of the main reasons of instability. Moreover, since biological analysis is listed in introduction as one of the possible applications, the synthetic blood plasma or urine solutions could be tested as well.

Author Response

Introduction should be improved with more detailed examples of microelectrode array analytical application. The importance of Ag/AgCl electrode fabrication instead of Ag (refer to screen printed electrodes) should be formulated more obvious.

Answer: Many thanks for bringing this issue up. The introduction is now revised with more examples and a better structure. The introduction of the analytical application of multielectrode arrays is now with more detailed examples. A brief description of wafer-level fabrication, i.e microelectronic process technologies, is now presented with a comparison of other technology (screen-printing). One of the challenges during the wafer-level sensor fabrication, i.e. fabrication of Ag/AgCl reference electrodes, is emphasized afterward with better-organized sentences.

According to Section 2.2, the two-electrode electrochemical cell was used in chlorination experiments. Why is it not the three-electrode electrochemical cell? Especially in potentiostatic mode, to be sure the measured potential is equal to desirable (applied).

Answer: Many thanks for noticing. A three-electrode electrochemical cell during the chlorination surely provides more reliable current/potential data. In practice, it was, however, quite challenging. The chlorination was carried out in a beaker. Even though the 6-inch wafer was diced into modules for chlorination. The module itself occupied a large space so that the space of the beaker barely holds a counter electrode and a stirrer. As the focus of our first stage study was to find out whether the electrochemical chlorination under metallic layers works at all, two-electrode electrochemical chlorination was carried out. In the future study, we will certainly consider this improvement.

The study looks incomplete. Despite the authors mentioning that it is the first stage of the investigation, the manuscript suffers from the lack of electrochemical data and/or experiments with improvements that the authors mentioned in the text (lines 333-343). As a suggestion: open circuit potential measurement should be enhanced with data and discussion for unmodified Ag layer and for solutions with lower KCl concentration or with another electrolyte (nitrate, sulphate, phosphate …) since chlorides diffusion is named one of the main reasons for instability. Moreover, since biological analysis is listed in the introduction as one of the possible applications, synthetic blood plasma or urine solutions could be tested as well.

Answer: Many thanks for the helpful input regarding the further characterization of the samples. These valuable suggestions will be considered in the next experimental campaign. As the electrode samples are not long-term stable after chlorination this requires a new series of chips. Due to the complex preparation of chip substrates, it is not possible to produce new samples in time for consideration of the results in this paper. We have thus also changed the introduction part of the paper to express more clearly the fundamental proof of concept nature of the work and the fact that this is a process-focused paper.

Reviewer 2 Report

This paper reported some results on the electrochemical growth of Ag/AgCl reference electrodes for miniaturized sensors. The topic is interesting and important. The authors partly succeeded in preparing Ag/AgCl electrodes. However, the reviewer considers that the main experiments and results are not sufficient in terms of both quality and quantity. In my opinion, it requires major revision before ready for publication. The points are listed below.

1) In Fig. 1, the authors presented six chips. However, the authors only focused on chip 1 and chip 6. Therefore, the authors should not present chip 2-5.

2) In Fig. 11, the potential started to change at most after 300 s. Why? Have you analyzed the measured samples such as SEM-EDS?

3) In conclusion, the authors mentioned “Large-scale Ag/AgCl…”, but the reviewer thinks it is not large scale but small scale.

4) In conclusion, the authors mentioned that “the long-term stability … depends highly on chip size design”, however, the reviewer could not see any differences between XS and XL in Fig. 12. Please show the evidence for this.

5) During AgCl formation from Ag, volume expansion must occur. It should be one of the reasons to cause crack formation in Fig 6. It should be also considered in the mechanism for AgCl formation in Fig. 10. The reviewer wants the authors to mention this point.

6) The authors used 3 M KCl to check stability. However, it may be better to use a lower concentration such as 1 M KCl for a miniaturized electrode because a high concentration of KCl may accelerate undesired AgCl formation. Have you ever tried such experiments?

7) In Fig. 7, what does (cs) mean? Please explain. Also, the arrows to show the distance (for example 2.081 µm in Fig. 7(c)) are difficult to see.

There are many typos in this manuscript. Especially, there were many instances where the basic rule of placing a space between units and numbers was not followed. (In Line 108, 109, 226, 228, 230, 236, 237, 241327, )

In abstract, “SEM & EDX” should be “SEM and EDX”.

In Fig. 2, “HCL/KCL” should be “HCl/KCl”.

In Line 359, “KCL” should be “KCl”.

In Fig. 8, “g)” should be “e)”.

In Line 358, “places, etc. grooves.” is not proper. Do you mean that “places, grooves, XX, etc.” or “places such as grooves”?

In Line 359, “saturated KCl” should be “3 M KCl” if Fig. 11 is correct.

Author Response

1) In Fig. 1, the authors presented six chips. However, the authors only focused on chip 1 and chip 6. Therefore, the authors should not present chip 2-5.

Answer: Thank you for pointing this out. Figure 1 is now changed with an illustration focusing on chips 1 and 6 with a detailed demonstration of the structures in cell XS and Cell XL.

2) In Fig. 11, the potential started to change at most after 300 s. Why? Have you analyzed the measured samples such as SEM-EDS?

Answer:  We believe that the discussions presented in section 3.3 have explained the results. In section 3.3 we have first shown the stability test (OCP measurements) results, then in subsections 3.3.1 and 3.3.2, we have discussed the main two factors that influence the stability of Ag/AgCl electrodes.

Instead of “SEM-EDS” we use the term “SEM-EDX”. EDX mapping analysis for chemical composition distribution was conducted after each chlorination. For the galvanostatic chlorination, both the EDX mapping analysis and the secondary electron images are presented in section 3.1.  Since for the sussesful potentiostatic chlorination only AgCl was detected, thus only secondary electron images were prese nted for potentiostatic chlorination in section 3.2.

3) In conclusion, the authors mentioned “Large-scale Ag/AgCl…”, but the reviewer thinks it is not large scale but small scale.

Answer: Thank you for bringing this issue up. We tend to emphasize the production scale is on the wafer level, which is a large scale. The original sentence is indeed misleading. Now the sentence is revised as follows

“Ag/AgCl reference electrodes with various geometric sizes between Ø 20 µm and Ø 200 µm at large-scale multielectrode arrays on wafer-level with metallic intermediate layers were successfully fabricated.”

4) In conclusion, the authors mentioned that “the long-term stability … depends highly on chip size design”, however, the reviewer could not see any differences between XS and XL in Fig. 12. Please show the evidence for this.

Answer: Figure 12 shows that there are always grain size differences between cell XS and cell XL under different chlorination conditions. The grain size was affected by the open area during deposition, which can be also described as chip size. Please allow us to quote the original text in the draft from line 329-line 344:

 “It is also natural to stress the importance of the AgCl grain size on long-term stability as the larger the grain size the more stable the AgCl layer can endure the external pressure. As shown in Figure 12, the duration of stability time of the reference electrode still holds a strong relationship with AgCl grain size regardless of the underneath porous AgCl layer and the existence of hollow gaps. That is, for long-term stability, the grain size played a much more important role. For the further performance improvement of the reference electrode, it is thus more important to maintain or even increase the AgCl grain size. Furthermore, the results from the two candidates cell XS and Cell XL in the chip have shown that it is very challenging to reach the same level of Ag chlorination for all the cells with different sizes without adding any reagents as the results shown in Figures 9 and 10. The large difference between the cells also led to a short stability time during the OCP measurement. The chip size has thus also an impact on the stability.     ” 

5) During AgCl formation from Ag, volume expansion must occur. It should be one of the reasons to cause crack formation in Fig 6. It should be also considered in the mechanism for AgCl formation in Fig. 10. The reviewer wants the authors to mention this point.

Answer: Thank you for bringing up this issue. Now the draft is revised mentioning this issue.

In line 178, we added, “The transformation of Ag to AgCl itself is a volume expansion procedure.”

In lines 287 and 288, we added, “The volume expansion during the AgCl formation is believed to contribute to the generation of nanopores and microchannels.”

6) The authors used 3 M KCl to check stability. However, it may be better to use a lower concentration such as 1 M KCl for a miniaturized electrode because a high concentration of KCl may accelerate undesired AgCl formation. Have you ever tried such experiments?

Answer: Thank you for bringing up this topic. Yes, we did consider trying a lower-concentration background solution. In fact, for the next step improvement, open circuit potential measurement for unmodified Ag layer and for solutions with lower KCl concentration or with another electrolyte (nitrate, sulphate, phosphate, etc) are considered. As the electrode samples are not long tern stable after chlorination, each OCP measurement requires a new series of chips. Due to the complex preparation of chip substrates, it is not possible to produce new samples in time for consideration of the results in this paper.

This paper presents our first-stage study of the wafer-level fabrication of the Ag/AgCl electrodes. Our first intention is to find out its commercial potential by comparing its performance with a commercial macro Ag/AgCl reference electrode, which has a 3M KCl background solution. Therefore, for the first run of OCP measurement, we don’t want an electrolyte concentration difference between the inner and outer background solution.

7) In Fig. 7, what does (cs) mean? Please explain. Also, the arrows to show the distance (for example 2.081 µm in Fig. 7(c)) are difficult to see.

Answer: Thank you for noticing. Cs stands for cross-section. Now Figure 7 is changed with notes explaining the meaning of the abbreviation cs. For a better illustration, the scales in the cross-section SE image are now bold.

Reviewer 3 Report

This study performed a new approach for design of multielectrode arrays based on Ag/AgCl reference electrodes with various layouts. It was well designed and organized, but it should be revised according to several following problems before publication.

1. The introduction is very poor. It cannot emphasize the importance of this research and its objectives is also not clear. Please expanding its contents and summarize more references to point out the problems and limitations of the recently published researches

2. Conclusion section is not well written. In this section, the author should concentrate sumarize the important results. it should not include discussions (line 352 - 356)

3. Please reorganize Figure 9 in a good shape and all figures must be labelled

4. Similarly, Figure 10 includes 4 images, but they are not numbered or labelled

5. Figure 1 includes 6 chips, but it was not specified and defined in the caption. 

6. In the results and discussions, I think that the authors only focus on listing the results. It completely lacks discusions for these results.

N/A

Author Response

  1. The introduction is very poor. It cannot emphasize the importance of this research and its objectives is also not clear. Please expanding its contents and summarize more references to point out the problems and limitations of the recently published researches

Answer: Many thanks for bringing this issue up. The introduction is now revised with more examples and a better structure. The introduction of the analytical application of multielectrode arrays is now with more detailed examples. A brief description of wafer-level fabrication, i.e microelectronic process technologies, is now presented with a comparison of other technology (screen-printing). One of the challenges during the wafer-level sensor fabrication, i.e. fabrication of Ag/AgCl reference electrodes, is emphasized afterward with better-organized sentences.

  1. Conclusion section is not well written. In this section, the author should concentrate sumarize the important results. it should not include discussions (line 352 - 356)

Answer: We believe that the potential growth mechanism is also an important conclusion, which is concluded from the discussion part in section 3.3.1.

  1. Please reorganize Figure 9 in a good shape and all figures must be labelled.

Answer: Thanks for bringing up this issue. Images in Figure 9 are now organized in better shape.

  1. Similarly, Figure 10 includes 4 images, but they are not numbered or labelled

Answer: Thanks for bringing up this issue. Now Figure 10 with the arrow of time on top demonstrates a clearer growth process. The three growth processes are now numbered underneath the images. The first image shows the nucleation at the very beginning. The second image shows the following generation of the first porous layer. The third and fourth images illustrate together the further generation of a compact layer out of a porous layer and the generation of a new porous layer, simutanously. We believe that the discussion combined with the current figure completes the description of the growth mechanism and the reaction of AgCl growth.

  1. Figure 1 includes 6 chips, but it was not specified and defined in the caption. 

Answer: Thanks for pointing out. Figure 1 is now changed with an illustration focusing on chips 1 and 6 with a detailed demonstration of the structures in cell XS and Cell XL.

  1. In the results and discussions, I think that the authors only focus on listing the results. It completely lacks discusions for these results.

Answer: We are sorry to hear that the reviewer has missed or misunderstood the discussion part in our manuscript. Each discussion was presented right after the description of the result. No separation was made in sections 3.1 and 3.2, which both belong to the galvanostatic chlorination of Ag. Now section 3 is rearranged with the title of each subsection presenting the discussion topic.

In section 3.1 we found out that galvanostatic chlorination led to strong corrosion of the cells. Thus, in subsection 3.1, we discussed the reason for the corrosion effect and improved the silver electrochemical deposition, which is a critical step that is worth mentioning in subsection 3.1.2.  In section 3.2 we have first listed all potentiostatic chlorination results. Based on the results we discussed the potential growth mechanism. In section 3.3 we presented the OCP measurement results. Based on the OCP measurement results, we discussed in section 3.3.1 the influence of nanopores and irregular formation of the first Ag nucleation on the stability of the Ag/AgCl electrode. In addition, in section 3.3.2 we discussed further the grain size and chip size effect on the stability of the Ag/AgCl electrode.

Round 2

Reviewer 1 Report

Authors have revised an introduction part and now it provides better drowning into the topic. Unfortunately, authors could not provide new experimental results listed in Reviewer’s recommendation “Due to the complex preparation of chip substrates”. Nevertheless, this version of manuscript looks quite appropriate as fundamental proof of concept.

Author Response

We would like to thank the reviewer for careful and thorough reading of this manuscript and for the thoughtful comments and constructive suggestions, which help to improve the quality of this manuscript and the future research. 

Reviewer 2 Report

1) Instead of “SEM-EDX,” the reviewer recommends the authors to use SEM-EDS or SEM-EDXS. Because EDS is an abbreviation of “Energy Dispersive X-ray Spectroscopy.” If you use EDX, it means Energy Dispersive X-ray. Of course, X-ray is not the name of analysis method. Thus, EDX is not the name of analysis method. Although the reviewer recognizes that many researchers misunderstand this point, you should use proper technical terms. If you really want to use EDX, it is better to use “EDX analysis”. Also, since you cannot map “X-ray”, the term “EDX mapping” is strange. It is better to use “elemental mapping by EDS”.

Anyway, the reviewer wanted to ask the reason for poor stability. Therefore, the reviewer recommends measuring SEM-EDXS after stability test to discuss the poor stability.

2) If the reviewer understands the results in Fig. 12 correctly, stability of XS and XL (red dashed line; although the reviewer could not understand why the authors use a dashed line instead of point-type marker) is very similar at each deposition condition (or it seems to be the same). Therefore, it seems that the stability does not depend on chip size. In addition, although grain size of samples deposited at 0.5 V and 1 V was similar, stability was different. Therefore, the grain size also did not affect the stability at these conditions. It implies that other factors should play major role in stability. Please explain this part clearly.

There are many typos in this manuscript. Especially, there were many instances where the basic rule of placing a space between units and numbers was not followed. (In Line 111, 1117, 118, 246, 247, 251) For example, 40s should be 40 s. Please be very careful for this point.

In Fig. 3, “HCL/KCL” should be “HCl/KCl”.

In Line 112, “KCL” should be “KCl”.

In Fig. 8, “g)” should be “e)”.

In Line 358, “places, etc. grooves.” is not proper. Do you mean that “places, grooves, XX, etc.” or “places such as grooves”?

In Line373, “saturated KCl” should be “3 M KCl” if Fig. 11 is correct.

There are many typos in this manuscript. Especially, there were many instances where the basic rule of placing a space between units and numbers was not followed. (In Line 111, 1117, 118, 246, 247, 251) For example, 40s should be 40 s. Please be very careful for this point.

In Fig. 3, “HCL/KCL” should be “HCl/KCl”.

In Line 112, “KCL” should be “KCl”.

In Fig. 8, “g)” should be “e)”.

In Line 358, “places, etc. grooves.” is not proper. Do you mean that “places, grooves, XX, etc.” or “places such as grooves”?

In Line373, “saturated KCl” should be “3 M KCl” if Fig. 11 is correct.

Author Response

We would like to thank the reviewer for careful and thorough reading of this manuscript and for the thoughtful comments and constructive suggestions, which help to improve the quality of this manuscript and the future research.

The point-to-point responses are listed as follows:

1) Instead of “SEM-EDX,” the reviewer recommends the authors to use SEM-EDS or SEM-EDXS. Because EDS is an abbreviation of “Energy Dispersive X-ray Spectroscopy.” If you use EDX, it means Energy Dispersive X-ray. Of course, X-ray is not the name of analysis method. Thus, EDX is not the name of analysis method. Although the reviewer recognizes that many researchers misunderstand this point, you should use proper technical terms. If you really want to use EDX, it is better to use “EDX analysis”. Also, since you cannot map “X-ray”, the term “EDX mapping” is strange. It is better to use “elemental mapping by EDS”.

Anyway, the reviewer wanted to ask the reason for poor stability. Therefore, the reviewer recommends measuring SEM-EDXS after stability test to discuss the poor stability.

Response: many thanks for pointing out the issues. Now the SEM-EDX is revised to ‘SEM-EDS’.

After the OCP measurement, the structure of the module1 is mainly destroyed and it is in practice impossible to be FIB cut and characterized by SEM-EDS.

(The module: a module is diced out from the 6-inch wafer, we have now mentioned in the manuscript)

We believe that the SEM-EDS characterization in parallel to the OCP measurement has gained sufficient understanding regarding the stability of the electrodes.

2) If the reviewer understands the results in Fig. 12 correctly, stability of XS and XL (red dashed line; although the reviewer could not understand why the authors use a dashed line instead of point-type marker) is very similar at each deposition condition (or it seems to be the same). Therefore, it seems that the stability does not depend on chip size. In addition, although grain size of samples deposited at 0.5 V and 1 V was similar, stability was different. Therefore, the grain size also did not affect the stability at these conditions. It implies that other factors should play major role in stability. Please explain this part clearly.

Response: thank you very much for bringing up this issue. We now understand the misleading point. The OCP measurement was carried out with the whole modules (totoal 6 chips in one module, with cell XS in chip 1 and Cell XL in chip 6). So we can have only information between different chlorination conditions. A detailed description of the OCP measurement is now added in section 2.4. The revised parts are as follows:

“The open circuit potential (OCP) measurement was carried out for a performance test of the fabricated Ag/AgCl electrode under the three potentiostatic chlorination conditions at 20°C. During the OCP measurements, 6-inch wafer was diced into modules, which containg a total 18 chips including cell XS and cell XL.”

If the module is further diced with each chip separated tested by the OCP measurement, we believe the result would be better. However, as the electrode samples are not long-term stable after chlorination in practice, and the dicing and bonding for single cell from the chip is time consuming (1-2 weeks), it is in practice quite challenged to measure the OCP stability test for each cell. However, the relationship between grain size and duration time still holds a strong relationship, even though there is no direct OCP measurement for each cell. In the manuscript, we have now revised section 3.3.2. The revised parts are as follows:

“Due to the practical challenge, there was no separated OCP measurement test for cell XS and cell XL under each chlorination condition. However, it can be seen in Figure 12 that, the duration of stability time of the reference electrode still holds a strong relationship with AgCl grain size as the larger the grain size in general in cell XS and cell XL, the longer the stability time is, regardless of the underneath porous AgCl layer and the existence of hollow gaps.”

There are many typos in this manuscript. Especially, there were many instances where the basic rule of placing a space between units and numbers was not followed. (In Line 111, 1117, 118, 246, 247, 251) For example, 40s should be 40 s. Please be very careful for this point.

In Fig. 3, “HCL/KCL” should be “HCl/KCl”.

In Line 112, “KCL” should be “KCl”.

In Fig. 8, “g)” should be “e)”.

In Line 358, “places, etc. grooves.” is not proper. Do you mean that “places, grooves, XX, etc.” or “places such as grooves”?

In Line373, “saturated KCl” should be “3 M KCl” if Fig. 11 is correct.

Response: Many thanks for finding out the typos. We have done another grammar and typo check and believe that now the draft is typo-free.

Reviewer 3 Report

It can be accepted for publication

N/A

Author Response

We would like to thank the reviewer for careful and thorough reading of this manuscript and for the thoughtful comments and constructive suggestions, which help to improve the quality of this manuscript and the future research.

We have done another grammar and typo check and believe that now the draft is typo-free.